**Data Availability Statement:** All relevant data are within the manuscript and its Supporting information files.

# The impact of using digitally-mediated social stories on the perceived competence and attitudes of parents and practitioners supporting children with autism

Louis John Camilleri[1,2]*, Katie Maras[1], Mark Brosnan[1]

**1** Centre for Applied Autism Research (CAAR), University of Bath, Bath, United Kingdom, **2** Department of Inclusion & Access to Learning, University of Malta, Malta, United Kingdom

* louisjohncamilleri@gmail.com

## Abstract

A Social Story (SS) is a highly acceptable and widely used intervention by the autism community. Yet, inconsistent implementation of the intervention is reported to be one of the causes of variability in terms of outcome research, particularly in a naturalistic context. This study aimed to (1) investigate whether digitally-mediated social stories (SSs) can improve competence in developing and delivering a SS and thus contribute towards improved implementation, and (2) investigate the impact of the digitally-mediated SS on attitudes towards the SS intervention. Ninety-three participants took an initial pre-engagement survey. Forty-eight of these participants also complete a post-engagement survey. A pre-post design was utilised with the participants who completed both surveys. These 48 participants were invited to develop a digitally-mediated SS with the aim of exploring how digitally-mediated SSs impacted perceived competence and attitudes. Post-engagement data was collected two weeks after the pre-engagement data. Outcomes of this study indicate that both perceived competence and attitudes improved after engaging with digitally-mediated SSs. It is concluded that digitally-mediated SS not only impacted the integrity of how the intervention was delivered but also the beliefs in the participants' capabilities to develop and deliver a SS. Digitally-mediated SS, thus, has the potential to effectively support development and delivery whilst also addressing challenges related to intervention implementation in a naturalistic context.

## Introduction

The Social Story (SS) intervention was developed by Carol Gray [1] specifically for children with autism (N.B. There is a lack of consensus on the terminology used to describe autism. In order to respect the diversity of views, both identity-first language as well as person-first language should be used [2]. However, in this paper, whilst acknowledging the diversity of views, for the purpose of stylistic consistency, person-first language–i.e., person with autism–is used

**Funding:** The author(s) received no specific funding for this work.

**Competing interests:** The authors have declared that no competing interests exist.

throughout). It is an intervention that comprises a highly structured and personalised narrative which describes a situation, skill, or concept in terms of relevant social cues, perspectives, and common responses in a specifically defined style and format [3]. The goal of social stories (SSs) is for a writer (also known as an "author"), usually a parent or a practitioner, to accurately and efficiently share social information with an individual (also referred to as the "audience") in a descriptive and informative manner [4, 5].

Originally SSs were developed and delivered solely in the form of written text or through the medium of print. Pictures were later introduced after a surge in interest in the intervention led to further research and a better understanding of SSs [6]. Several methods for the administration of the intervention have been used subsequently, which include: digital tablets [7] social robots [8], computer software [9] and also Wearable Immersive Virtual Reality (WIVR) technology [10].

SS is described as a highly acceptable intervention by practitioners in the field of autism [11]. It is also one of the most frequently used interventions by parents of children with autism [12, 13]. Attitudes towards the intervention are reported to be positive [14, 15], which could be a result of the high social validity reported by parents and also by people with autism themselves [16]. Also, this could even be a consequence of the apparent simplicity [17] and/or versatility of the intervention.

There have been a large number of studies illustrating the positive effects of social stories on a number of behaviours. These include anxiety [18], challenging behaviour [19], understanding emotions [20] developing social skills [21, 22], coping with new situations [23] understanding sexuality [24] and increasing task engagement [25], amongst others. These "goals" have been grouped in four categories by Kokina et al. [26]: (i) reduction of inappropriate behaviours, (ii) improvement in social skills, (iii) teach academic or functional skills, and (iv) assist in transition/novel situation/reduce anxiety. However, systematic and meta-analyses highlight great variability in terms of effectiveness and outcomes of the intervention across all four areas [11, 26–30].

Reasons for such variability have been suggested by several researchers. One of the reasons which could be contributing to this variability is the goal which the story is targeting. SS that target reducing negative behaviours are reported to be the most used in research [31], whilst other SS goals, such as managing transitions are reported to only make up around 9% of the literature [26]. Constantin et al. [32] suggest that the audience's poor or inconsistent understanding of SS could also be another factor which is contributing to this variability. Comprehension checks are recommended by Gray's original criteria [1], where it is recommended that authors always "Plan for Comprehension". Constantin et al. [32] argue that comprehension checks are often skipped by parents and practitioners, which means it is unknown whether the child's understanding of the situation has improved or not. This issue, which is brought forward by Constantin et al. [32] is related to poor procedural integrity of the intervention.

## Procedural integrity

Procedural integrity—also known as procedural fidelity or treatment integrity–is essential for empirical testing of intervention efficacy and dissemination of evidence-based practices [33]. A variety of definitions and conceptual models have been proposed to define procedural integrity, which is considered a wide-ranging concept [34]. The definition used in this study is that proposed by Perepletchikova et al. [35], who define Procedural integrity in terms of three components [11, 35, 36]: agent competence (i.e., the knowledge base and skill that the individual who is implementing an intervention exhibits), treatment adherence (i.e., the reliable use of the procedures as specified a priori), and treatment differentiation (i.e., the extent that the

intervention is reliably distinguished from another intervention). In their synthesis of the literature, Test et al. [29] and Bucholz [3] suggest that poor procedural integrity could be impacting SS intervention effectiveness, and thus contributing towards the variability in outcomes for SS.

## Variability and the use of technology

Digital technology can aid the reduction of variability, and the use of digitally mediated interventions and supports in the field of autism has been on the rise over the past decade [37]. With the development of smartphone/tablet technology and smartphone/tablet apps, their use by children with autism and their families has become increasingly popular [38]. However, Kim et al. (2018) warn that research on digitally-mediated interventions has not kept pace with the volume apps available to support autism. In fact, from a total of 700 smartphone applications, Kim et al. [39] found that only 0.6% of them had any empirical evidence to support their usage. Chen [40] postulates that research regarding the impact and effectiveness of digital technologies for children with autism are inconclusive. Nevertheless, innovative technologies carry great promise. Chen [40] argues that service users and providers still need to understand how to take advantage of the affordance of this technology to provide the best possible support for individuals with autism, their families and professionals who support them (see Riga et al. [41]). Thus, generally, there seems to be agreement that, whilst digital support is a promising area for autism, it still requires further research [7, 39, 42].

More recently, Hanrahan et al. [43] utilised digitally-mediated SS to address the issue of variability in the development and delivery of SS (see also Min et al. [44] and Vandermeer et al. [45]). Using a digital tablet, the authors evidenced the effectiveness of digitally-mediated SS employing a Randomised Control Trial design. When compared with the outcomes of a control group, digitally-mediated SS were found to be effective in producing a beneficial behavioural change in children with autism [43]. Similarly, in a study by Smith et al. [46], teachers used a digital tablet application to develop and deliver personalised digitally-mediated SSs with children with autism over a 4-week intervention period. Outcomes of this study indicated that teachers were able to carry out the intervention with a high degree of procedural integrity. Furthermore, improvements in child understanding as well as decreased anxiety.

In another study, conducted by Smith et al. [47], digitally-mediated SS were used by summer camp teachers to support ten children with autism through the transition to the summer camp. Outcomes of this study were again indicative of appropriate procedural integrity and resulted in significant improvements in appropriate behaviours and reductions in inappropriate behaviours. Consistent with Gray's criteria, these studies identified a specific goal for the SS intervention and measured intervention success by analysing practitioners' ratings of the child's closeness to the target goal [48].

## Competence and procedural integrity

Thus, with features such as ensuring goal-setting and comprehension checks, digital technologies can support parents and practitioners to develop and deliver SS to children with autism consistent with Gray's criteria. However, the effectiveness and integrity of the intervention will still depend on the competence of the author of the story [49]. "Competence" here is defined as the extent to which an author has the knowledge and skill required to deliver a treatment to the standard needed for it to achieve its expected effects [50]. This suggests that competence is directly related to procedural integrity, as it would be difficult to ensure that interventions are implemented as intended without a good knowledge of that intervention. Procedural integrity tends to be higher in interventions led by researchers who are experts in the field of SS, and lower in studies where development and delivery of the intervention is carried out by

individuals will less competence in delivering the intervention [51, 52]. Furthermore, improved competence, resulting from direct teaching/coaching of authors (i.e., an individual who is developing and delivering the intervention), tends to lead to positive outcomes in naturalistic contexts where SS intervention is developed and delivered by individuals such as parents and practitioners, rather than by researchers (see Acar et al. [15] and Olçay-Gül et al. [53]).

### Aims of the study

With a view to improving procedural integrity, through a pre-post research design, this study aimed to investigate if a digitally mediated SS improved competence ratings of "authors" who are developing and delivering SS in a naturalistic setting i.e., in a context that is outside research confines, where confounding variables cannot be controlled. It also aimed to explore factors that could predict competence outcomes, and also investigate the impact of a digitally mediated SS on the author's attitude towards the intervention. A final analysis also looked at the different goals identified by authors who used the app to create a SS. The various goals were categorised in pre-defined groups, as described by Kokina et al. [26], in order to examine if there were any differences between categories of SS goals.

### The SOFA-app

SSs in this study were digitally mediated through the use of the SOFA-app (Stories Online for Autism). The SOFA-app is an online application, for smart devices (Android/iOS smartphones or tablets), through which social stories can be developed and delivered. It is an app that was co-developed with the autism community [54] that is aimed towards helping authors create and deliver SS reliably and effectively. The SOFA-app (https://SOFA-app.org) is free-of-charge, on the Google-Play store and Apple store.

### Relevance of this study

This study is relevant to the autism community as it contributes towards answering the question of high variability in SS research. Furthermore, this study also contributes towards the evidence, or lack of, that is currently available to substantiate the effectiveness of digital supports for autism interventions [37].

Moreover, as a result of the COVID-19 pandemic, digital platforms for interventions are increasing in popularity [55]. Recently, practitioners are more inclined to suggest technological interventions to families that need more accessible services [56]; not only because of the accessibility of digital supports, but also because of the reported affinity with technology children and adolescents with autism are reported to exhibit [57, 58]. Caria et al. [59] hypothesise that this could be "due to the predictable and repeatable nature of technology that can create controlled environments, and which thus appeals to those (particularly children with autism) who feel relieved by stability and routine" (p.1735).

## Method

### Participants

Ninety-three participants (68 practitioners and 25 parents) completed the baseline measures and were given access to the application. Subsequently, 48 of the initial 93 participants (34 practitioners and 14 parents) used the digitally-mediated SS and completed the outcomes measures. This represents a completion rate of 52%.

The parents that participated in this study all indicated that they were parents of children with autism. The practitioners that participated were Psychologists (coming from the Educational, Clinical, Health, and Counselling fields), Speech-Language Pathologists (SLP), Occupational Therapists (OTs), Teachers, Learning Support Educators (LSEs), Teaching Assistants (TAs), Inclusion Coordinators (INCOs), and Special Education Needs Coordinators (SENCOs) and all worked with children with autism.

Participants were recruited through a non-probability purposive sampling strategy. The recruitment was carried out by emailing invitations to an administrative contact at a programme, in Malta, that provides services for children with autism and their parents. Recruitment was also carried out online through a Massive Open Online Course (MOOC) that focused on evidence-based practices for autism in education. The participants' characteristics, their experience with SS, as well as their proficiency with digital mobile technology, were identified and are presented in Table 1.

## Procedures

The participants were invited to complete a pre-engagement baseline questionnaire evaluating perceived competence and attitude towards social stories. The link for this baseline questionnaire, together with comprehensive information about the study, was made available to

**Table 1. Participants' demographics.**

|  | Completed Baseline Only[1] | | Completed Baseline & Outcome[2] | | All participants[3] | |
|---|---|---|---|---|---|---|
|  | *n* | P(%) | *n* | P(%) | *n* | P(%) |
| Role |  |  |  |  |  |  |
| Parents | 11 | 24.4 | 14 | 41.2 | 25 | 26.9 |
| Practitioners | 34 | 75.6 | 34 | 58.8 | 68 | 73.1 |
| Total | 45 | 100 | 48 | 100 | 93 | 100 |
| Experience with SS |  |  |  |  |  |  |
| Extensive experience | 35 | 77.8 | 35 | 72.9 | 70 | 75.3 |
| Little/no experience | 10 | 22.2 | 13 | 27.1 | 23 | 24.7 |
| Participant's Age Range |  |  |  |  |  |  |
| 18 to 25 | 2 | 4.4 | 3 | 6.2 | 5 | 5.4 |
| 26 to 35 | 13 | 28.9 | 16 | 33.3 | 29 | 31.2 |
| 35 to 45 | 17 | 37.8 | 22 | 45.8 | 39 | 41.9 |
| 46 to 55 | 11 | 24.5 | 5 | 10.4 | 16 | 17.2 |
| 56 to 70 | 2 | 4.4 | 2 | 4.3 | 4 | 4.3 |
| Participant's Country of residence |  |  |  |  |  |  |
| Malta | 25 | 55.6 | 38 | 79.2 | 63 | 67.7 |
| UK | 20 | 44.4 | 10 | 20.8 | 30 | 32.3 |
| Perception of one's proficiency at using an electronic device (mobile or digital tablet) |  |  |  |  |  |  |
| Proficient | NA[4] | | 45 | 93.8 | NA[4] | |
| Not proficient |  | | 3 | 6.2 |  | |

P = *Percentage*, NA = Not applicable.

[1] Participants who completed only the baseline measures (n = 45).

[2] Participants who completed the baseline and outcome measures (n = 48).

[3] Participants from [1] + participants from [2] (N = 93).

[4] No baseline or outcome data was collected. Perception of one's proficiency at using an electronic device was gathered only for participants who completed outcome measures for competence and attitude.

potential participants through email, and by posting information together with a link on the MOOC online forum. All of the study procedures were completed using Qualtrics® Software (www.qualtrics.com).

Participants who completed the baseline questionnaire were invited to download a SS app (SOFA-app) on their smartphone or tablet and were asked to complete two tasks: With a particular child with autism in mind; (i) identify a goal for a target behaviour to be addressed by a SS, and (ii) develop an appropriate SS. Practitioners and parents were given two weeks to get acquainted with the app and complete the assigned tasks. Participants were then invited to complete the measures again (i.e., complete a post-engagement survey).

## Measures

Initial questions were aimed at establishing the level of experience the participants had with developing and delivering SS. Participants who had experience with both developing and delivering SS were grouped in the "Extensive experience" category. Participants who had experience only with delivering or only with developing a SS were grouped in the "Little/No experience" category. Participants who had never developed nor delivered a SS where also grouped in the "Little/No experience" category. The level of confidence in one's ability to develop and deliver SS was gauged by asking participants to state their level of agreement, on a 5-point Likert, for the following statements: (i) "How confident are you in your ability to write an effective social story?" and (ii) "How confident are you in your ability to effectively use a social story with a child?".

Subsequently, the participants were invited to complete a set of baseline questions which were presented in the form of statements. All participants were asked to indicate their level of agreement/level of confidence on a 5-point Likert scale ranging from strongly agree/very confident (5) to strongly disagree/very "unconfident" (1). Participants were asked questions aimed to explore: their confidence rating in their ability to develop and deliver a SS (Confidence Scale), their attitude towards SS intervention (Attitude Scale), and their perceived level of competence in their ability to develop and deliver a SS (Competence Scale). The 48 participants who complete the outcome measures were also invited to rate the perception of their proficiency (proficient or not proficient) of using their particular electronic device (smartphone/ digital tablet) and also to rate their experience (User Experience) of using the application to digitally-mediate the intervention. The latter was calculated by computing the mean scores of participants on four questions aimed at gathering information about the users' experience with the application (see S1 Appendix).

The items used in the Attitude Scale were adapted from the Intervention Rating Profile– 15 [60] and the Abbreviated Acceptability Rating Profile [61]. These measures were developed to explore acceptability of the intervention.

Competence was operationally defined in terms of two aspects [62]: (i) having the knowledge of the intervention (measured through questions C2, C3 & C6) and (ii) having the preparedness to deliver the treatment as intended (measured through questions C1, C4 and C5). Furthermore, the questions comprising the Competence Scale aimed to measure the participants' perceived competence from a "limited-domain" perspective [62]. Through this conceptual lens, competence is seen as "treatment-specific" and thus is concerned about the specific intervention, i.e., the SS intervention. Thus, questions were intended to elicit the participants' perceived understanding of goal setting (C4), sentence construction (C3 & C5), SS treatment complexity (C1), SS treatment specificity (C6), and perceived skill required to develop a SS (C2).

## Data analysis

For the participants who completed both the baseline and outcome measures (n = 48), Kokina et al.'s [26] coding categories (reduction of inappropriate behaviours; improvement in social skills; teach academic or functional skills; assist in transition/novel situation/reduce anxiety) were used to classify the targets identified by the authors. i.e., the goal of the digitally-mediated SS.

For each participant, we averaged ratings of agreement over 2 "confidence" items, 6 "attitude" items, and 6 "competence" items, and computed summary scores for each scale (see Table 3). For participants who completed the outcomes measures, rating on 4 "user experience" (UE) items were average and computed into one variable. All of the scales have good internal consistency, with a Cronbach Alpha coefficient of 0.75 (Confidence), 0.85 (Attitude), 0.75 (Competence) and 0.82 (User Experience). Baseline and outcome attitude and competence ratings were recorded and analysed. The degree of change registered on competence and attitude scales after using the application was also computed.

The Shapiro-Wilks test for normality was used to detect departures from normality on all computed measures. When analysing all of the baseline data (N = 93), normality assumptions were violated for baseline measures (baseline confidence, baseline competence and baseline attitude). When analysing data obtained from participants who completed both baseline and outcome measures (n = 48), the normality assumption was not violated for baseline competence ($p = .082$) and outcome competence ($p = .247$). Change in outcome competence rating was trending towards normality ($p = .035$). This trend was confirmed by visual inspection of a histogram. Baseline and outcome attitude scores violated the normality assumption.

Thus, in cases where the normality assumption was not violated, parametric tests (paired-sample t-test, Pearson product-moment correlation and simple linear regression) were utilised. This was the case when analysing competence ratings for data obtained from participants who completed both baseline and outcome measures. Non-parametric tests (Wilcoxon signed-rank, Kruskal-Wallis test and Spearman's rank-order correlation) were utilised when analysing attitude ratings since the data obtained from attitude measures all failed to meet the normality condition. In instances when differences identified in data was found to be statistically significant ($p < .05$), effect sizes were calculated by using Cohen's d measure for the magnitude of experimental effect [63].

## Ethical considerations

This study was approved by the University of Bath Psychology Research Ethics Committee (PREC, 19–309) and by the University of Malta's Faculty Research Ethics Committee (FREC, 2393_25072019). All procedures performed in this study were in accordance with the 1964 Helsinki declaration and its later amendments. Data were collected through the use of online questionnaires. In these questionnaires, the participants were informed about the purpose of the study and consent was gathered online. Assurance was given to participants that their involvement in the study was strictly voluntary.

## Results

Descriptive statistics that illustrate the basic features of the data in this study are presented in Tables 2 and 3. Table 2 provides a summary of the mean, standard deviation, and median ratings for baseline data gathered with all 93 participants. Table 3 provides similar summary data to Table 2, but only for the participants who completed both baseline and outcome measures (n = 48).

**Table 2. Descriptive statistics of baseline measures from all participants (N = 93).**

|  | Total | | Parents | | Practitioners | | Extensive Experience[5] | | Little/no experience[6] | |
|---|---|---|---|---|---|---|---|---|---|---|
|  | M (SD) | Mdn | M (SD) | Mdn | M (SD) | Mdn | M (SD) | Mdn | M (SD) | Mdn |
| Baseline confidence | 3.56 (0.63) | 4.00 | 3.52 (0.66) | 4.00 | 3.58 (0.62) | 4.00 | 3.66 (0.59) | 4.00 | 3.15 (0.60) | 3.00 |
| Baseline competence | 3.74 (0.67) | 3.83 | 3.41 (0.73) | 3.67 | 3.86 (0.61) | 4.00 | 3.91 (0.56) | 4.00 | 3.22 (0.73) | 3.33 |
| Baseline attitude | 4.55 (0.45) | 4.67 | 4.23 (0.58) | 4.17 | 4.66 (0.32) | 4.67 | 4.62 (0.38) | 4.67 | 4.33 (0.58) | 4.33 |

M = Median; SD = Standard Deviation; Mdn = Median

[5] Participants who reported to have extensive experience with developing and delivering SS intervention.

[6] Participants who reported to have little to no experience with developing and delivering SS intervention.

## Baseline measures

There was a statistically significant positive correlation, with a medium effect size, between baseline competence and baseline confidence ($r_{s} = .34$, n = 87, $p < .001$), and between baseline competence and baseline attitude ($r_{s} = .34$, n = 93, $p < .001$). The relationship between baseline attitude and baseline confidence was small ($r_{s} = .16$, n = 93, $p > .05$) and not statistically significant.

**Table 3. Descriptive statistics of measures from participants who complete post-engagement survey (n = 48).**

|  | Total | | Parents | | Practitioners | | Extensive experience | | Little/no experience | |
|---|---|---|---|---|---|---|---|---|---|---|
|  | M (SD) | Mdn | M (SD) | Mdn | M (SD) | Mdn | M (SD) | Mdn | M (SD) | Mdn |
| Competence[7] | | | | | | | | | | |
| Baseline Competence rating | 3.68 (0.76) | 3.75 | 3.30 (0.75) | 3.59 | 3.84 (.71) | 4.00 | 3.88 (0.64) | 3.83 | 3.11 (0.78) | 2.83 |
| Outcome Competence ratings | 4.10 (0.50) | 4.17 | 4.05 (0.56) | 4.09 | 4.12 (.47) | 4.17 | 4.09 (0.47) | 4.17 | 4.11 (0.59) | 4.17 |
| Change* in Competence ratings | 0.42 (.79) | 0.25 | 0.75 (0.97) | 0.33 | .28 (.68) | 0.25 | 0.20 (.66) | 0.17 | 1.00 (0.87) | 0.83 |
| Attitude[8] | | | | | | | | | | |
| Baseline Attitude ratings | 4.54 (.43) | 4.67 | 4.27 (0.57) | 4.17 | 4.65 (0.31) | 4.67 | 4.59 (0.36) | 4.67 | 4.37 (0.57) | 4.50 |
| Outcome Attitude rating | 4.71 (0.37) | 4.83 | 4.71 (0.32) | 4.75 | 4.71 (0.39) | 4.92 | 4.70 (0.40) | 4.83 | 4.74 (0.28) | 4.83 |
| Change* in Attitude rating | 0.18 (0.46) | 0.17 | 0.44 (0.45) | 0.33 | 0.07 (0.42) | 0.00 | 0.10 (.41) | 0.17 | 0.37 (0.52) | 0.33 |
| Baseline Confidence[9] | 3.60 (0.65) | 4.00 | 3.55 (0.64) | 4.00 | 3.61 (0.66) | 4.00 | 3.63 (0.62) | 4.00 | 3.43 (0.79) | 4.00 |
| User Experience (UE) ratings[10] | 4.11 (0.67) | 4.25 | 4.20 (0.71) | 4.63 | 4.08 (0.67) | 4.25 | 4.16 (0.54) | 4.25 | 4.00 (0.97) | 4.50 |

M = Mean; SD = Standard Deviation; Mdn = Median

* Change is referring to the mean of the difference between outcome and baseline ratings.

[7] Extent to which an author has the knowledge and skill required to deliver a SS intervention to the standard needed for it to achieve its expected effects.

[8] Disposition with regards to SS.

[9] Belief in one's ability to deliver SS intervention appropriately.

[10] Perception of the "ease of use" of the application used for digital-mediation.

**How does role and experience impact baseline measures?.** A statistically significant difference was found in baseline competence ratings of experienced (n = 70) and the less experienced participants (n = 23), U = 371.0, z = -3.87, p < .001, r = .40. A statistically significant difference was also present in baseline attitude (U = 568.5, z = -2.14, p = .033, r = .22) and confidence ratings (U = 320.5, z = -3.15, p = .002, r = .33) of experienced participants and less experienced participants. As expected, those with more experience reported more confidence, competence and positive attitudes at baseline.

The impact of role on baseline measures, i.e., the differences in baseline ratings between parents (n = 25) and practitioners (n = 68) was also analysed. A statistically significant difference was registered in baseline competence (U = 556.5, z = -2.5, *p* = .011, r = .26) and baseline attitude (U = 452.5, z = -3.49, *p* < .001, r = .36) between the two group of participants, but not in baseline confidence (U = 667.0, z = -.277, p = NS). This indicates that practitioners and parents did not report a significant difference in confidence ratings. However, practitioners reported higher competence and attitude scores at baseline.

## Outcome measures

Forty-eight participants (34 Practitioners and 14 parents), from the initial 93, completed the baseline as well as the outcome measures. No significant differences on confidence, competence and attitude measures were obtained between participants who completed baseline measures only, and participants who completed both baseline and outcome measures (all *ps* >0.85. The means and medians for the outcome measures are in Table 3.

**Competence.** *Does competence change after using digitally-mediated SS*? Participants' (n = 48) competence ratings prior to using the application (M = 3.68, SD = .76) increased after using digitally-mediated SS (M = 4.09, SD = 0.49). The result was indicative of a statistically significant mean increase of 0.42 (SD = .79), BCa 95% CI [0.19, 0.65], t(47) = 3.65, p = .003. A Cohen's d of .65 indicated a medium effect size.

*How do baseline and outcome competence relate to each other*? There was a statistically significant, strong negative correlation between the two variables, *r*(46) = -.79, *p* < .001, with high levels of baseline competence ratings resulting in a lower change in outcome competence ratings.

*How does baseline confidence relate to change in competence*? There was a statistically significant, negative correlation, with medium effect size, between the variables, with the resulting degree of change in competence decreasing as baseline competence increased: $r_s$(40) = -.33, *p* = .032.

*How does experience with SS impact change in competence*? The participants with extensive experience (n = 35) obtained a mean increase of .205 (*SD* = .112). However this increase was not statistically significant, BCa 95% CI [.004, .434] t(34) = 1.84, *p* = .074 The participants with little to no experience (n = 13) obtained a statistically significant mean increase of 1.01 (SD = .233), BCa 95% CI [.564, 1.451], *t(12)* = 4.148, *p* = .001. A Cohen's *d* of 1.45 indicated a large effect size.

The change in competence was higher in the less experienced group (*M* = 1.00, *SD* = 0.87) than in the group with extensive experience (*M* = .20, *SD* = 0.66). This was statistically significant, 95% CI [-1.27, -.32], *t*(46) = -3.40, *p* = .001, with a Cohen's *d* of 1.03, indicative of a large effect size.

*How does role impact change in competence*? Practitioners' competence ratings prior to using the application (M = 3.835, SD = .713) increased after using the application to develop a SS (*M* = 4.119, *SD* = 0.475). The result was indicative of a statistically significant mean increase of 0.284, 95% CI [0.046, 0.523], t(33) = 2.43, p = .021. A Cohen's *d* of .47 indicated a trending

medium effect size. Parents' initial competence ratings ($M$ = 3.30, SD = .75) also improved after using digitally-mediated SS ($M$ = 4.05, SD = 0.56). This difference, .75, BCa 95% CI [.28, 1.22] was also statistically significant $t$(13) = 2.90, $p$ = .020, and represented a large effect size, Cohen's $d$ = 1.13.

The change in competence, as a result of using digitally-mediated SS, was higher for parents ($M$ = .75, $SD$ = 0.97) than for practitioners ($M$ = .28, $SD$ = .68). However, this result was not statistically significant (95% CI [.13., 1.06], $t$(18.6) = -1.638, $p$ = .065).

*Does "user experience" (UE) relate to change in competence*? There was no statistically significant correlation between UE and change in competence, $r_s$(46) = .125, p = .40.

**Attitude.** *Does attitude change with use*? Of the 48 participants recruited for the study, the application elicited an increase in attitude ratings in 25 participants post-intervention, whereas 7 participants' attitude ratings decreased post-intervention. There was a median increase in attitude ratings of .167 from baseline (Mdn = 4.67) to outcome (Mdn = 4.83) ratings. This difference was statistically significant, z = -3.07, p = .002, r = .31.

*How do baseline and outcome attitude relate to each other*? There was a statistically significant, strong negative correlation between baseline attitude rating and degree of change in attitude ratings after using the app, $r_s$(46) = -.66, $p < .001$, with higher baseline confidence ratings resulting in a lower change in attitude after using the application.

*How does baseline confidence relate to change in attitude*? There was no statistically significant correlation between the variables, $r_s$(40) = -.134, $p$ = .397.

*How does experience with SS impact change in attitude*? The app elicited a post-intervention increase in attitude ratings in 18 of the experienced participants and 7 of the less experienced participants. 5 experienced participants and 2 less experienced participants registered a decrease in attitude. Attitude ratings of 12 experienced participants and 4 with little to no experience remained unchanged. There was a median increase in attitude ratings from baseline (Mdn = 4.67) to outcome (Mdn = 4.83) ratings for the experienced group. This difference, .16, was statistically significant, z = -2.228, $p$ = .026, r = .27. The attitude ratings of participants with little to no experience also improved after using the digitally-mediated SS. The results of the signed-rank test indicated that this difference, .33, was also statistically significant, z = -2.322, $p$ = .020, $r$ = .46.

A difference in attitude rating was found between groups of differing levels of experience (Mdn = .16), with participants with less experience registering more attitude improvement (Mdn = .33) when compared to participants with more experience (Mdn = .17). However, this difference was not statistically significant, U = 183.50, z = -1.05, $p$ = .296.

*How does role impact change in attitude*? An increase in attitude ratings was registered in fourteen of the practitioners, whereas six practitioners' attitude ratings decreased. There was a median increase in attitude ratings of 0.245 from baseline (Mdn = 4.670) to outcome (Mdn = 4.915) ratings, but this difference was not statistically significant, z = -1.501, p = .133. Parents' attitude ratings also improved after using the digitally-mediated SS. This difference, 0.33 was statistically significant, z = -2.95, p = .003.

*Does "user experience" (UE) relate to attitude change*? There was no statistically significant correlation between UE and change in attitude, $r_s$(46) = .047, $p$ = .749.

*How does change in outcome competence relate to change in outcome attitude ratings*? There was a statistically significant, positive correlation with a medium effect size between the two variables, $r_s$(46) = .416, $p$ = .003, with high competence ratings associated with higher attitude ratings towards the intervention.

**Goals of SS developed.** The goals of the SS developed by parents and by practitioners were classified in terms of the 4 main categories of SS goals described by Kokina & Kern's [26]: (1) reduction of inappropriate behaviours, (2) improvement in social skills, (3) teach

academic or functional skills, and (4) assist in transition/novel situation/reduce anxiety. The researchers independently categorised 47 of the goals/targets identified by the parents and practitioners and obtained 87% interrater agreement. Cohen's κ was also run to determine if there was agreement between the researchers' judgement. There was strong agreement between author's judgements, κ = .885, 95% CI [.777, .993], $p < .001$. Any discrepancies in ratings were resolved via discussion resulting in 100% agreement. Finally, it was agreed that "reduction of inappropriate behaviours" represented 18.8% of goals; "improving social skills" represented 22.9%; "teaching academic or functional skills" represents 20.8%; and "assisting in transition/novel situation/reduce anxiety" represented 35.4% of goals identified. There were no significant differences between competence-change or attitude-change for any of the 4 groups (all $p$s > 0.061).

## Discussion & implications

Social stories (SS) are a popular, acceptable and widely used intervention for children with autism [12, 13, 64, 65]. Attitudes to SS are positive in both parents and practitioners, although variability in competence, with regards to implementing the intervention, has raised questions regarding effectiveness [15, 51–53]. It has been proposed that digitally-mediated social stories can support the competence of the story writer (author) and improve intervention integrity.

This study found that using digitally-mediated social stories significantly improved perceived competence and attitudes for both parents and practitioners with higher and lower levels of previous experience with the intervention. Importantly, digitally-mediated social stories were found to be beneficial outside of the research context, with parents and practitioners writing social stories in a naturalistic context–in some instances for the first time. Those with the least levels of competence and positive attitudes, perceived the greatest improvement in their competence and positive attitudes. The findings are therefore consistent with previous research which indicates that digitally mediated social stories can reduce the variability in the development of the stories and the delivery of the intervention, and improve effectiveness [43, 46, 47]. This study suggests that this may extend to parents and practitioners with a wide range of levels of previous experience.

This study was one of the first to investigate the impact of digitally-mediated SS in a naturalistic context, specifically from the author's perspectives. Unlike previous studies where digitally-mediated SS intervention were used in a naturalistic setting [46], in this study researchers did not author the stories. Rather, parents and practitioners were "tasked" with the authoring of the stories. Thus, for the first time, this study shows that parents and practitioners can write the stories themselves.

Baseline and outcome data collected in this study also indicates that the higher the participants' perceived competence ratings, the more positive the attitude towards SS. Furthermore, outcome data indicates an increase in attitude ratings after using the digitally-mediated SS. Whilst experienced and less experienced participants both registered an increase in attitude, results indicated that there was no meaningful difference between the attitude increase of the two groups. This indicates that experience did not meaningfully impact attitude increase. On the other hand, a statistically significant increase in attitude was registered in parents but not in practitioners. This could indicate that role does impact change in attitude towards SS. Outcomes of this study also indicate that competence ratings improved in both experienced and less experienced participants. Nonetheless, participants with less experience registered more improvement than "experienced" participants. Also, participants with high levels of baseline competence and confidence ratings reported a smaller degree of change in outcome competence ratings; i.e., participants who reported

higher baseline competence and confidence also reported the least change in improvement whilst still registering an increase.

Moreover, outcomes of this study also indicate that competence ratings improved in both parents and practitioners who participated in this study. Furthermore, it seems that that parents' competence ratings improved more than practitioners' competence ratings. Such improvements in competence also seemed to be dependent on the participant's levels of experience with the intervention prior to using the application. Thus, this implies that competence–in terms of goal setting, sentence construction, and ratio of descriptive to coaching sentences–can be improved, to various degrees, through digital mediation. This finding is noteworthy, especially in light of the lack of research on the implementation of interventions for children with autism in naturalistic settings. Improved competence and procedural integrity can contribute greatly towards improved effectiveness of SS interventions in a naturalistic setting [50], and also towards decreasing barriers in implementation. To overcome these barriers Kasari and Smith [66] postulate that key components of interventions have to be clear whilst also allowing for flexibility. The issue of flexibility is particularly important for the area of autism, since heterogeneity of presentation is a hallmark of this neurodevelopmental condition [67]. Thus, the clinical features and cognitive differences characteristic of autism imply that an individual with autism requires a specific and personalised intervention [68]. This places parents and practitioners in an ideal and unique position to develop and deliver specific interventions since they tend to be most knowledgeable about an individuals' unique characteristics. Furthermore, the adaptability of SS intervention also lends itself well to this goal. Actually, in this study, the personalised nature of the intervention was exemplified by how each participant in the study was able to identify a goal which was pertinent to the child/individual they had in mind, adapt the intervention to that individual's needs, and all whilst being conscious of integrity requirements. This means that SS interventions could be delivered with a higher degree of procedural integrity by individuals who know the "service users" best.

The findings are therefore consistent with previous studies, such as those carried out by Smith et al., [46, 47] and by Constantin et al. [32], who suggest that digitally-mediated SS could be effective in supporting children with autism. This could be achieved by utilising digital-mediation to maximise procedural integrity by improving an "author's" competence to develop and deliver a SS. Furthermore, similar to Smith et al., [47], this study suggests that digital mediation could contribute towards better accessibility to the intervention for authors, and also reduce consumption of time and effort for both developing and delivering the intervention. Digital mediation of the intervention may therefore address the variability in effectiveness that has been identified in previous studies [11, 26–30].

Such findings are encouraging especially when considering the positive attitude towards the intervention reported in various studies. This study also suggests that the use of a digitally mediated SS could further improve positive attitudes towards the intervention, even when the intervention is used in a naturalistic context. Consequently, this could encourage practitioners and parents to increasingly utilise digitally mediated SS in their everyday interactions with children with autism providing greater intervention consistency across contexts. The accessibility and consistency of use across contexts could potentially lead, in turn, to greater effectiveness of the intervention. Future research can explore this potential through applying the methodology in this study to children with autism, and monitoring the impact on the child's understanding, or behaviour. Also, future research could focus on obtaining qualitative data which could also elucidate further the participant's experience of the intervention as well as the medium used to deliver the intervention, for example by explicitly comparing digital and non-digital SS.

## Limitations

This study explored authors' experience of the app, and a limitation is that the stories were not used with children. However, it important to establish for future research that the authors are able to develop the intervention. Another limitation is that the study assessed perceived competence. It was not possible to independently assess the SS that had been written. Again, this can be a feature of future research.

From the 93 participants that showed interest in this study, only 48 finished both the baseline and the outcomes questionnaires, which is another limitation of the study. Furthermore, the low number of parents who participated, compared to practitioners, is a further limitation. Nevertheless, it is difficult to control the number of responses obtained in a study, especially in a population such as that of parents of children with autism who report high burden, psychological distress, and lower social support [69].

Furthermore, it is important to note that 94% of the participants that completed the required activities, together with all the outcome measures, reported that they were proficient in their ability to use digital technology. Only 6% described themselves as "not proficient". The participants' extremely high level of proficiency could be revealing of a positive bias towards technology, which could have in turn skewed both their attitude and competence ratings. Also, recruitment invited participation to take part in a study of SS, and it is likely that those with an interest in the intervention were more likely to participate. This will reduce the generalisability of the findings. Though it is interesting to note that some participants had no experience with SS.

No data on the participants' gender or ethnicity was gathered in this study. However, there currently is no research that indicates that these factors have any impact on the variables and processes that were investigated in this study.

The research design utilised meant that the outcomes could not be compared to other media. Comparison of digital mediation with non-digital media could have added a comparison in changes in competence and attitude outcomes resulting from different modes of delivery. Comparison of competence and attitude outcomes as a result of using other applications, other than SOFA, would have also aided in understanding if the extent to which results of this study were specific to this app. Finally, future research could focus on obtaining data about children's or adolescents' perceptions of SS intervention as well as digitally mediated SS. A question which could be addressed is if SS could be developed by the children/adolescents with autism themselves and if they could be supported to do this through digital technology.

## Conclusion

Overall, the findings from the present study indicate that using digitally mediated SS could improve, to various degrees, parents and practitioners' competence and attitudes towards developing the intervention for children with autism. This is the case for those with higher and lower levels of experience with the intervention, with those with lower levels potentially benefitting the most from using digitally mediated SSs.

## Supporting information

**S1 Table. Practitioners, baseline & outcomes (n = 34), descriptive statistics.**
(DOCX)

**S2 Table. Parents, baseline & outcomes (n = 14), descriptive statistics.**
(DOCX)

**S1 Appendix. Questionnaire items.**
(DOCX)

## Author Contributions

**Conceptualization:** Louis John Camilleri, Mark Brosnan.

**Data curation:** Louis John Camilleri.

**Formal analysis:** Louis John Camilleri.

**Methodology:** Louis John Camilleri, Mark Brosnan.

**Supervision:** Katie Maras, Mark Brosnan.

**Writing – original draft:** Louis John Camilleri.

**Writing – review & editing:** Louis John Camilleri, Katie Maras, Mark Brosnan.

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
