## [Decision Letter · Decision Letter 0]

22 Jul 2021

PONE-D-21-16546

The Impact of Using Digitally-Mediated Social Stories on The Perceived Competence and Attitudes of Parents and Practitioners Supporting Children with Autism.

PLOS ONE

Dear Dr. Camilleri,

Thank you for submitting your manuscript to PLOS ONE. After careful consideration, we feel that it has merit but does not fully meet PLOS ONE’s publication criteria as it currently stands. Therefore, we invite you to submit a revised version of the manuscript that addresses the points raised during the review process.

We look forward to receiving your revised manuscript.

Kind regards,

Mingming Zhou, Ph.D.

Academic Editor

PLOS ONE

Journal Requirements:

2. Thank you for including your ethics statement: "All procedures performed in this study were in accordance with the ethical standards of the institutional and/or national research committee (University of Bath Psychology Research Ethics Committee – PREC and University of Malta Faculty of Education Research Ethics Committee), and with the 1964 Helsinki declaration and its later amendments or comparable ethical standards."

a) Please amend your current ethics statement to confirm that your named institutional review board or ethics committee specifically approved this study.

b) Please provide additional details regarding participant consent. In the ethics statement in the Methods and online submission information, please ensure that you have specified (1) whether consent was informed and (2) what type you obtained (for instance, written or verbal, and if verbal, how it was documented and witnessed). If your study included minors, state whether you obtained consent from parents or guardians. If the need for consent was waived by the ethics committee, please include this information.

3. Peer review at PLOS ONE is not double-blinded (https://journals.plos.org/plosone/s/editorial-and-peer-review-process). For this reason, authors should include in the revised manuscript all the information removed for blind review.

Reviewers' comments:

Reviewer's Responses to Questions

**Comments to the Author**

1. Is the manuscript technically sound, and do the data support the conclusions?

Reviewer #1: Yes

Reviewer #2: Partly

Reviewer #3: Partly

2. Has the statistical analysis been performed appropriately and rigorously? 

Reviewer #1: Yes

Reviewer #2: Yes

Reviewer #3: Yes

3. Have the authors made all data underlying the findings in their manuscript fully available?

Reviewer #1: Yes

Reviewer #2: Yes

Reviewer #3: Yes

4. Is the manuscript presented in an intelligible fashion and written in standard English?

Reviewer #1: Yes

Reviewer #2: Yes

Reviewer #3: Yes

5. Review Comments to the Author

Reviewer #1: Broadly, the noble goals or contributions of the research can be explained in a straightforward manner. The title already represents the overall picture of the article.

The phenomenon of Digitally-Mediated Social Stories on the Perceived Competence and Attitudes of Parents and Practitioners Supporting Children with Autism as the main figure in the article, can be strengthened again regarding the emergence of the phenomenon.

At the end of the introduction, it is necessary to strengthen the issue of Digitally-mediated social stories and how they can be used in the future for "the new normal" on autism methods. What makes SSs more interesting and urgent to discuss? Can it also be linked to the IoT generation or digital natives? on the other hand, do autistic children also have generations like children in general?

Triggers or lighters regarding the current existence of SSs and their correlation to multidimensional models of autism also need to be sharpened as an effort to lead readers' minds about the direction of the research being carried out.

Reviewer #2: The article presents The Impact of Using digitally-mediated Social Stories on The Perceived Competence

and Attitudes of Parents and Practitioners Supporting Children with Autism.

The article is well written, and sufficient data is provided. However, the study's overall conclusion is difficult to generalize with the children with Autism, as the parents participated and not directly the children. In the case of levels of ASD, every child is different from others within the same level. So the supported data was mandatory about the level of autistic kids (High functioning, low functioning, age); this is a significant limitation. Plus, there should be a reference group as well to compare the results. The details and reason for choosing this specific statistical analysis should also be mentioned. With the current data, the general conclusion is too optimistic.

Reviewer #3: This paper made a subjective self-report investigate on the implementation of using digitally-mediated social story, especially SOFA-app. Both pre and post surveys were collected and analyzed. The result shows that using digitally-mediated SS significantly improved perceived competence and attitudes for both parents and practitioners. For details:

1. In the analysis of the degree of change on competence and attitude in pre and post engagements, whether there exists any difference on the baseline measures using 93 participants (including 45 people who did not complete the post-engagement survey) and baseline using 48 participants who completed both pre and post surveys?

2. As for competence, 6 ratings of competence items are averaged, since COMP1 and COMP6 are reverse scored, is it appropriate to directly average the scores of the 1&6 capability items equally?

3. Objective evaluations on competence are suggested to included, such as how many sentences were constructed on each goal and how much time was spent.

4. Besides currently self-reported rating, manually ratings by other professionals may also include to assess the quality of social story descriptions, such as clarity.

5. Suggest to add a brief description about the process of how to use SOFA-app, I am not sure whether there exists any difference in the ways that different participates use this tool.

6. PLOS authors have the option to publish the peer review history of their article (what does this mean?). If published, this will include your full peer review and any attached files.

Reviewer #1: No

Reviewer #2: No

Reviewer #3: No

---

## [Author Response · Author response to Decision Letter 0]

12 Aug 2021

• Issues pertaining to journal requirements:

a. Amend your current ethics statement to confirm that your named institutional review board or ethics committee specifically approved this study. 

b. Provide additional details regarding participant consent.

Authors’ reply:

Points a & b, above, have been addressed. In the methods section titled “Ethical considerations” the following has been added:

Ethical considerations

This study was approved by the University of Bath Psychology Research Ethics Committee (PREC, 19-309) and by the University of Malta’s Faculty Research Ethics Committee (FREC, 2393_25072019). All procedures performed in this study were in accordance with the 1964 Helsinki declaration and its later amendments. Data were collected through the use of online questionnaires. In these questionnaires, participants were informed about the purpose of the study and their consent was gathered online. Assurance was given to participants that their involvement in the study was strictly voluntary. 

• Reviewer #1 wrote:

At the end of the introduction, it is necessary to strengthen the issue of Digitally-mediated social stories and how they can be used in the future for "the new normal" on autism methods. What makes SSs more interesting and urgent to discuss? Can it also be linked to the IoT generation or digital natives? on the other hand, do autistic children also have generations like children in general? Triggers or lighters regarding the current existence of SSs and their correlation to multidimensional models of autism also need to be sharpened as an effort to lead readers' minds about the direction of the research being carried out.

Authors’ reply:

We have added the following text at the end of the introduction:

Relevance of this study

This study is relevant to the autism community as it contributes towards answering the question of high variability in SS research. Furthermore, this study also contributes towards the evidence, or lack of, that is currently available to substantiate the effectiveness of digital supports for autism interventions (Zervogianni et al., 2020). 

Moreover, as a result of the COVID-19 pandemic, digital platforms for interventions are increasing in popularity (Doenyas, 2021). Recently, practitioners are more inclined to suggest technological interventions to families that need more accessible services (Heng et al., 2021); not only because of the accessibility of digital supports, but also because of the reported affinity with technology children and adolescents with autism are reported to exhibit (Hedges et al., 2018; Laurie et al., 2019). Caria et al. (2018) hypothesise that this could be “due to the predictable and repeatable nature of technology that can create controlled environments, and which thus appeals to those [particularly children with autism] who feel relieved by stability and routine” (p. 1735).

With regards to the question posed by Reviewer #1: “Can it also be linked to the IoT generation or digital natives? on the other hand, do autistic children also have generations like children in general?” We feel that the question is indeed a pertinent one. However, answering that question goes beyond the scope of this study. Furthermore, the study’s participants were parents and practitioners, and not the children/adolescents with autism themselves. The questions that were asked to this specific sample were not aimed towards answering the reviewer’s question specifically, but whether digitally mediated SS improved competence ratings of “authors” who are developing and delivering SS in a naturalistic setting.

• Reviewer #2 wrote: 

The study's overall conclusion is difficult to generalize with the children with Autism, as the parents participated and not directly the children. In the case of levels of ASD, every child is different from others within the same level. So, the supported data was mandatory about the level of autistic kids (High functioning, low functioning, age); this is a significant limitation. Plus, there should be a reference group as well to compare the results. The details and reason for choosing this specific statistical analysis should also be mentioned. With the current data, the general conclusion is too optimistic.

Authors’ reply:

The study focuses on parents and practitioners’ perceived competence and attitudes. Thus, whilst the reviewer is right to state that “every child is different from others within the same level”, this has little to no bearing on parents’ and practitioners’ competence and attitudes towards social stories. The study aimed to investigate if a digitally mediated SS improved competence ratings of “authors” who are developing and delivering SS in a naturalistic setting. Thus, asking about the children’s “levels” would have raised ethical concerns as it would have been data/information which is surplus to requirements. 

The reviewer also requests information about the statistical methods used: “The details and reason for choosing this specific statistical analysis should also be mentioned “. We feel that this information is already substantial and the reasons for using the methods of analysis have been already discussed in the Data Analysis section of the Methodology:

In cases where the normality assumption was not violated, parametric tests (paired-sample t-test, Pearson product-moment correlation and simple linear regression) were utilised. This was the case when analysing competence ratings for data obtained from participants who completed both baseline and outcome measures. Non-parametric tests (Wilcoxon signed-rank, Kruskal-Wallis test and Spearman's rank-order correlation) were utilised when analysing attitude ratings since the data obtained from attitude measures all failed to meet the normality condition. In instances when differences identified in data was found to be statistically significant (p < .05), effect sizes were calculated by using Cohen’s d measure for the magnitude of experimental effect (Cohen, 1988).

Reviewer #2 also stated that the conclusion is too optimistic. To address this, we have made some changes to the conclusion:

Overall, the findings from the present study indicate that using digitally mediated SS could improve, to various degrees, parents and practitioners’ competence and attitudes towards developing the intervention for children with autism. This is the case for those with higher and lower levels of experience with the intervention, with those with lower levels potentially benefitting the most from using digitally mediated SSs.

• Reviewer #3 (no.1) wrote: 

1. In the analysis of the degree of change on competence and attitude in pre and post engagements, whether there exists any difference on the baseline measures using 93 participants (including 45 people who did not complete the post-engagement survey) and baseline using 48 participants who completed both pre and post surveys?

Authors’ reply:

This information is included under the heading Baseline Measures in the Results section:

48 participants (34 Practitioners and 14 parents), from the initial 93, completed the baseline as well as the outcome measures. No significant differences on confidence, competence and attitude measures were obtained between participants who completed baseline measures only, and participants who completed both baseline and outcome measures (all p > .05).

• Reviewer #3 (no.2) wrote: 

2. As for competence, 6 ratings of competence items are averaged, since COMP1 and COMP6 are reverse scored, is it appropriate to directly average the scores of the 1&6 capability items equally?

Authors’ reply:

The questions the reviewer is referring to are the following:

COMP1 - I believe that social stories are difficult for me to write or create.

COMP6 - Social stories are tools used for entertaining children and not an intervention.

Both of these questions can be considered “reversed-polarity items”. They have been incorporated to see to acquiescence response bias. On the Competence composite score (where all competence items were average), the higher the score obtained, the higher the perceived competence of the participant. If these two items were not reverse scored, the competence score would have not represented an accurate measure of the overall competence items. 

• Reviewer #3 (no. 3 & 4) wrote: 

3. Objective evaluations on competence are suggested to included, such as how many sentences were constructed on each goal and how much time was spent.

4. Besides currently self-reported rating, manually ratings by other professionals may also include to assess the quality of social story descriptions, such as clarity.

Authors’ reply:

We agree that these are good suggestions. However, the analysis of the stories was not part of the research design as the focus was to investigate perceived competence and attitude. Thus, the stories produced were not recorded. 

• Reviewer #3 (no.4) wrote: 

5. Suggest to add a brief description about the process of how to use SOFA-app, I am not sure whether there exists any difference in the ways that different participates use this tool.

Authors’ reply:

Including information about the process of “how to use SOFA-app” could be useful for readers to understand further. However, given the space constraints, this information can instead be obtained by following the link https://SOFA-app.org, which can be found under the subheading titled The SOFA-app.

---

## [Decision Letter · Decision Letter 1]

9 Dec 2021

PONE-D-21-16546R1The Impact of Using Digitally-Mediated Social Stories on The Perceived Competence and Attitudes of Parents and Practitioners Supporting Children with Autism.PLOS ONE

Dear Dr. Camilleri,

Thank you for submitting your manuscript to PLOS ONE. After careful consideration, we feel that it has merit but does not fully meet PLOS ONE’s publication criteria as it currently stands. Therefore, we invite you to submit a revised version of the manuscript that addresses the points raised during the review process.

We look forward to receiving your revised manuscript.

Kind regards,

Mingming Zhou, Ph.D.

Academic Editor

PLOS ONE

Journal Requirements:

Reviewers' comments:

Reviewer's Responses to Questions

**Comments to the Author**

1. If the authors have adequately addressed your comments raised in a previous round of review and you feel that this manuscript is now acceptable for publication, you may indicate that here to bypass the “Comments to the Author” section, enter your conflict of interest statement in the “Confidential to Editor” section, and submit your "Accept" recommendation.

Reviewer #4: All comments have been addressed

Reviewer #5: All comments have been addressed

2. Is the manuscript technically sound, and do the data support the conclusions?

Reviewer #4: Yes

Reviewer #5: Yes

3. Has the statistical analysis been performed appropriately and rigorously? 

Reviewer #4: N/A

Reviewer #5: Yes

4. Have the authors made all data underlying the findings in their manuscript fully available?

Reviewer #4: Yes

Reviewer #5: Yes

5. Is the manuscript presented in an intelligible fashion and written in standard English?

Reviewer #4: Yes

Reviewer #5: Yes

6. Review Comments to the Author

Reviewer #4: The authors performed all the required revisions that were requested by the previous round of review.

Reviewer #5: The manuscript is important and well written. I hereby suggest to add the following reference (if found relevant) in introduction/review of literature section: Riga, A., Ioannidi, V., & Papayiannis, N. (2021). SOCIAL STORIES AND DIGITAL LITERACY PRACTICES FOR INCLUSIVE EDUCATION. European Journal of Special Education Research, 7(2). doi:http://dx.doi.org/10.46827/ejse.v7i2.3773

7. PLOS authors have the option to publish the peer review history of their article (what does this mean?). If published, this will include your full peer review and any attached files.

Reviewer #4: No

Reviewer #5: No

---

## [Author Response · Author response to Decision Letter 1]

26 Dec 2021

• Reviewer #5 wrote:

The manuscript is important and well written. I hereby suggest adding the following reference (if found relevant) in the introduction/review of literature section: Riga, A., Ioannidi, V., & Papayiannis, N. (2021). SOCIAL STORIES AND DIGITAL LITERACY PRACTICES FOR INCLUSIVE EDUCATION. European Journal of Special Education Research, 7(2). doi:http://dx.doi.org/10.46827/ejse.v7i2.3773

Authors’ reply: 

Riga et al.’s (2021) reference is a theoretical piece that highlights how social stories (SSs), particularly digitally-mediated SSs, are useful for the autism community. 

As recommended by Reviewer #5, this reference has been included in the introduction/literature review (under the subheading “Variability and the use of technology”). 

We hope that this sees to the points raised by the reviewer. 

We look forward to your feedback. 

Best regards

THE AUTHORS

26th December 2021

---

## [Editor Report · Decision Letter 2]

30 Dec 2021

The Impact of Using Digitally-Mediated Social Stories on The Perceived Competence and Attitudes of Parents and Practitioners Supporting Children with Autism.

PONE-D-21-16546R2

Dear Dr. Camilleri,

We’re pleased to inform you that your manuscript has been judged scientifically suitable for publication and will be formally accepted for publication once it meets all outstanding technical requirements.

Kind regards,

Robert Didden

Academic Editor

PLOS ONE
---

## [Editor Report · Acceptance letter]

5 Jan 2022

PONE-D-21-16546R2 

The Impact of Using Digitally-Mediated Social Stories on the Perceived Competence and Attitudes of Parents and Practitioners Supporting Children with Autism. 

Dear Dr. Camilleri:

I'm pleased to inform you that your manuscript has been deemed suitable for publication in PLOS ONE. Congratulations! Your manuscript is now with our production department. 

Kind regards, 

on behalf of

Professor Robert Didden 

Academic Editor

PLOS ONE